# Integrating Reservoir Operations and Flood Modeling with HEC-RAS 2D

**Matthew Garcia *** 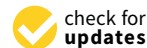**, Andrew Juan and Philip Bedient**

Department of Civil and Environmental Engineering, Rice University, Houston, TX 77005, USA;
andrew.juan@rice.edu (A.J.); bedient@rice.edu (P.B.)

**\*** Correspondence: msg6@rice.edu; Tel.: +1-713-348-4221

**Abstract:** Current free to use models developed by the United States Army Corps of Engineers (USACE) perform unique functions (e.g., hydrology, hydraulics, reservoir operations, and flood impact analysis) that are widely used in numerous studies and applications. These models are commonly set up in a framework that is limited to point source connections, which is problematic in regions with flat topography and complex hydrodynamics. The separate models need to be integrally linked and jointly considered for accurate risk communication and decision-making, especially during major storm events. Recently, Hurricane Harvey (2017) exposed the shortcomings of the existing framework in West Harris County, TX, where an insufficient understanding of potential flood risk and impacts contributed to the extensive flood damages sustained in the region. This work illustrates the possibility of using a single hydraulic model, HEC-RAS 2D, to perform all hydrologic, hydraulic, and reservoir operations modeling necessary for accurate flood impact assessments. Implications of this study include a simplification of the entire flood impact analysis, which could help future flood risk communication and emergency planning.

**Keywords:** urban reservoir operations; urban flood modeling; flood warning; reservoir modeling

## 1. Introduction

The U.S. Army Corps of Engineers' (USACE) reservoir operations (e.g., flood storage, water supply, and flow regulation) are typically simulated using the Hydrologic Engineering Center—Reservoir System Simulation (HEC-ResSim) software [1]. ResSim is often used in conjunction with other HEC software packages within the Corps Water Management System (CWMS) platform. CWMS and its publicly available equivalent, watershed analysis tool (WAT), are designed to integrate multiple model processes including hydrology (HMS), reservoir operations (ResSim), hydraulics (RAS), and flood impact (FIA). While useful, setting up and running CWMS or WAT properly is challenging because of a seamless integration across all linked models being required. Moreover, CWMS and WAT are limited to point source connections, which is problematic in flat-slopped regions with frequent high-intensity storms and complex hydrodynamics. Although other modeling software exists that integrate these computations (e.g., MIKE, TUFLOW, InfoWorks ICM, etc.), the prohibitively high costs of these systems, along with the requirement of most U.S. municipalities to use the official HEC software, make their use infeasible for many regions. Harris County, Texas, encountered the limitations of the current HEC system when the region was devastated by Hurricane Harvey in August 2017 [2]. Within the region, the Addicks and Barker reservoir system and their vicinity were among the most severely impacted. Because of the record-breaking rainfall, the reservoirs were forced to release massive outlet flows accompanied by uncontrolled spillway runoff that caused extensive flood damage [3]. Until recently, the potential flood extent and local impact caused by emergency reservoir releases in major storms such as Harvey were poorly understood. Updating the CWMS models preemptively for major

flood events is a massive undertaking due to the problems mentioned above. To address this issue while still maintaining the use of HEC software, all the necessary flood depth assessment processes were integrated in a single model. This study demonstrates that complex hydrodynamics, reservoir operations, and flood impact modeling can be computed while accounting for their interdependencies with a single hydraulic model, HEC-RAS 2D. This model integration will simplify the preemptive analysis of the flood impacts from reservoir operation changes and mitigate the devastating effects seen in the Addicks and Barker region from Hurricane Harvey in Houston and other regions in the future.

## 2. Methodology

### 2.1. Study Area

Harris County, TX has an average ground slope of 0.02% and an average annual rainfall of 136 cm since 2001 [4]. In the past two decades, rapid urbanization (sprawl from the Houston metro) has exacerbated the region's flood vulnerability as it shifts away from costal prairie and rice fields. This study's area of interest is 1526 km$^2$ in West Harris County, encompassing several watersheds: Upper Cypress Creek, Addicks and Barker Reservoirs, Buffalo Bayou, and a portion of White Oak Bayou (Figure 1). During major storms, extensive out-of-bank inundation alongside inter-basin overflows from Upper Cypress Creek into Addicks have been known to occur. Addicks and Barker, a pair of dry reservoirs consisting of earthen levees with concrete spillways at their ends, discharge into Buffalo Bayou which runs eastward through downtown Houston where it meets White Oak Bayou. While recent studies have evaluated individual watersheds within this West Houston region [5–7], this study intends to holistically evaluate the complex hydrodynamics, reservoir operations, and flood impacts as one system. Another major distinction from a typical hydrodynamic model run of this region is the length of the simulation. Since reservoir operations happen on the scale of weeks to months rather than days, the hindcast of Hurricane Harvey was run from 24 August 2017 through 20 October 2017.

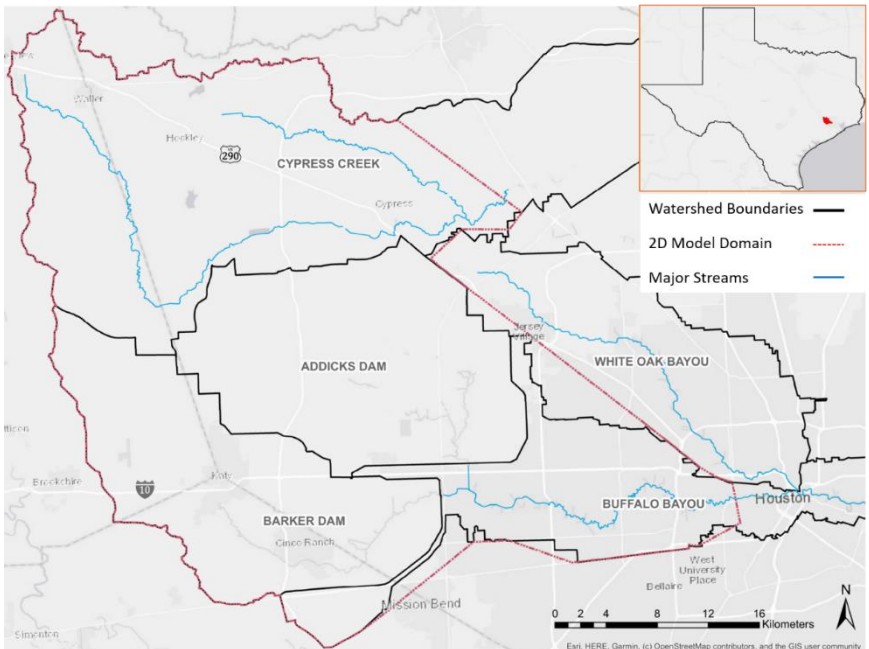

**Figure 1.** Overview of West Houston, TX watersheds.

### 2.2. Hydrodynamic and Reservoir Operation Modeling (HEC-RAS 2D) Overview and Setup

This study uses the 2D component of HEC-RAS (available in version 5.0.0 and more recent versions) [8] for hydrodynamic, reservoir operation, and floodplain modeling. HEC-RAS 2D generates

a mesh with properties derived from terrain and land use/land cover (LULC) datasets to represent the study area. Stormwater is introduced into the model domain by setting boundary condition inflows at specific locations, applying rainfall over the mesh, or both. HEC-RAS 2D then computes the flow, flood depth, water surface elevation, and velocity for every cell as the water is conveyed throughout the domain to the exit boundary conditions. Recent versions of HEC-RAS (i.e., version 5.0.5 and newer) also include limited scripting capabilities (rule based operations) for structures in 2D domains, allowing for certain flood control procedures such as sluice gate operations to be coded into the model.

Figure 2 shows the 2D model domain, which is developed in HEC-RAS using LIDAR terrain data from 2018 and consists of over 181,000 cells. It has a base mesh resolution of 122 m (400 ft), with higher resolution cells defined along levees (61 m or 200 ft) and channels (15 m or 50 ft). The Manning's roughness values for the domain are associated with land cover data from the 2016 National Land Cover Database (NLCD) [9], with USACE records of the reservoir operations (gate opening, pool elevation, rate of rise, etc.) roughness values being adapted from Kalyanapu et al. 2009 [10]. To simulate Harvey, spatially averaged net rainfall data from NOAA's Multi-Radar/Multi-Sensor System (MRMS) [11] raster was applied to the entire domain. Net rainfall was applied to account for infiltration in the model domain since HEC-RAS 2D currently does not compute infiltration loss.

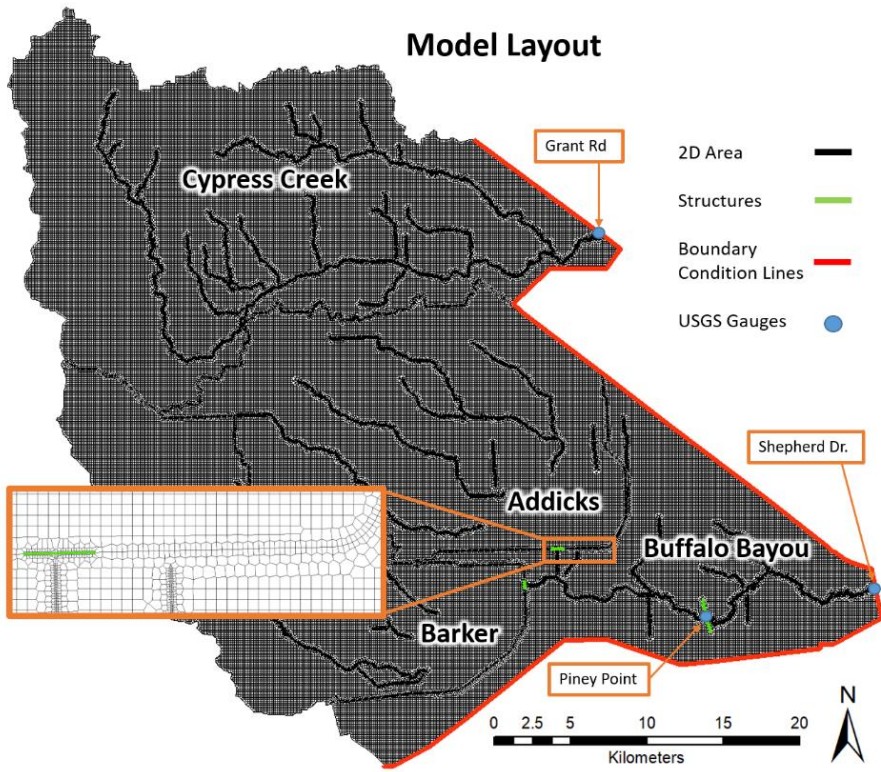

**Figure 2.** HEC-RAS 2D model domain, boundary conditions, and gauge locations.

The Addicks and Barker reservoirs were each modeled with a weir and gated culvert matching the existing structures' geo-referenced location, cross-sectional area, and levee heights (Figure 3). Since the USACE records of the reservoir operations (e.g. gate opening, pool elevation, and rate of rise) in Harvey significantly deviated from the Water Control Manual (WCM) [12] that the scripted gates were based on, the recorded opening data from the USACE was also modeled and used for the calibration. The two runs presented only differ by the method of operation for the reservoir gates and shows the difference between the WCM and the human operation of the structures. More information detailing the gate operation scripts (Section S.1), a GUI image of the code (Figure S1), and a version to copy (Section S.2) are presented in the Supplemental Materials. Finally, the downstream boundary conditions for the

model were rating curves at the corresponding United States Geological Survey (USGS) stream gauge locations [13] along the edge of the model domain and normal depth for overland areas.

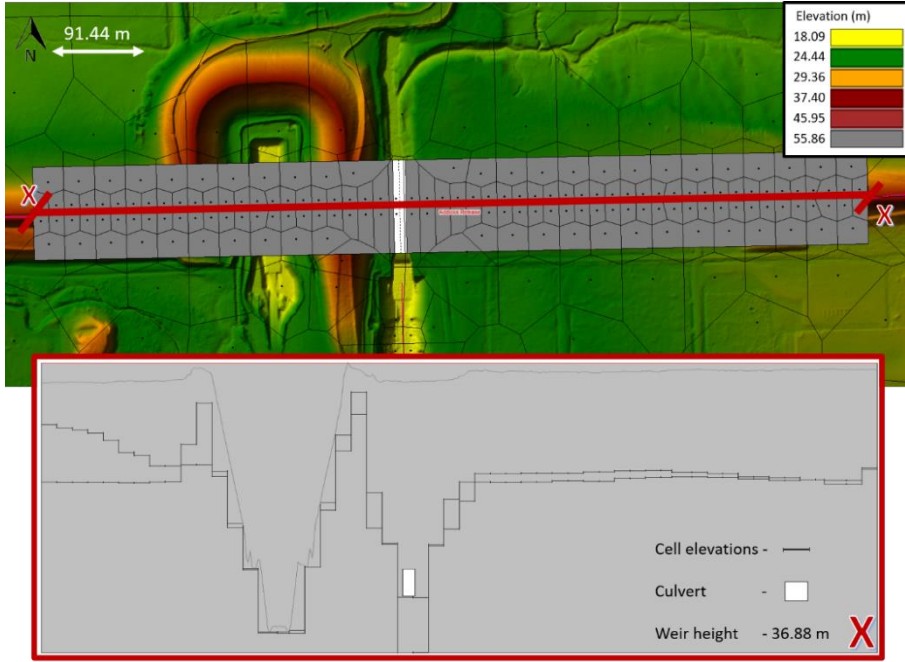

**Figure 3.** Plan and cross-sectional view of modeled outlet structure for Addicks Reservoir.

## 3. Results and Discussion

The RAS 2D model of the observed operations (labeled "Modeled" below) accurately computed the spillway overflows, the outlets' tailwater conditions, and the local fluvial and pluvial flooding which is evident from the comparisons at the reservoir pools (Figure 4), at the channel gauges (Figure 5), and across the modeled floodplain (Figure 6a,b, and Table 1). These figures also show the major distinction between observed operations and the coded WCM operations (labeled "Coded" below). The average Nash-Sutcliffe efficiency (NSE) [14] of the Modeled run for the gauges excluding Cypress Creek was 0.97 with an average root-mean-square error (RSME) of 0.18. The Coded run clearly performed much worse having a different operation schedule than the observed, but it shows the similarity of the two runs at the Cypress Creek gauge which was unaffected by the reservoir operations. The key result for the Coded run is the Piney Point gauge shown in Figure 5. This gauge is used for the determination of releases in the WCM and a distinct drop off around 9 September 2017 12:00 can be seen in the observed and Coded runs. This drop off is due to the reservoir pool levels dropping below the range for emergency releases. Because of the much milder release schedule of the actual event, the jump on 9 September 2017 in the USGS record is much less pronounced than the more aggressive releases from the Coded run. As a result of this difference, the observed jump is lost to attenuation by the time it reaches Shepherd Drive where it can still be seen in the Coded run. These similarities in timing and overall shape show that the coded operations are performing the functions to mimic the WCM as desired. While the Modeled run results generally matched the observed records very well for regions at and downstream of the reservoirs, the lack of spatially-distributed rainfall in the domain led to an overestimation of the volumes in Cypress Creek (recorded total precipitation was 22 cm less in the northwest portion of the domain than the southwest during the storm). To address the rainfall distribution limitation seen in the Cypress Creek results in the future, new work is currently being conducted with the setup and linking of multiple smaller 2D domains, each with its individual hyetographs to account for the spatial and temporal variability in the rainfall.

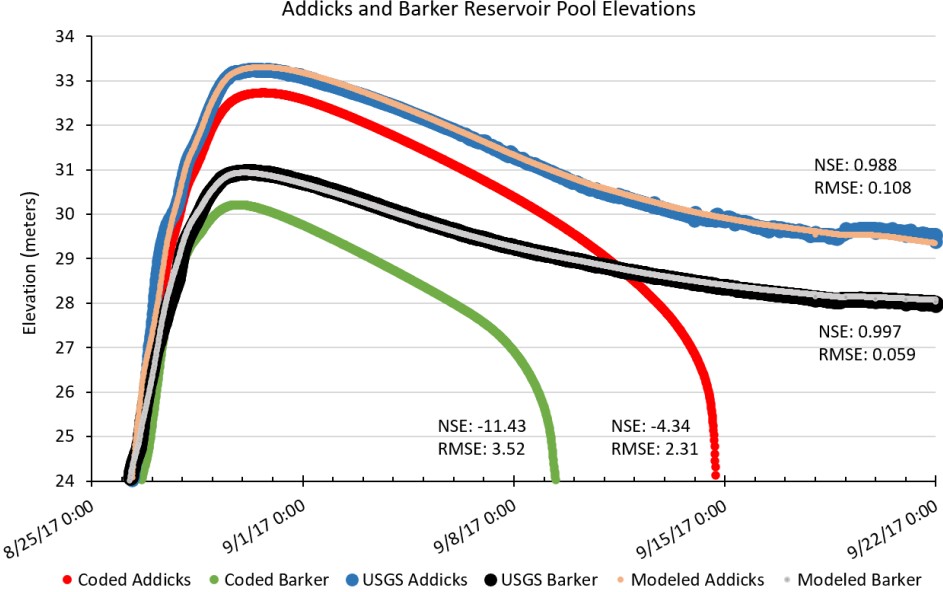

**Figure 4.** Addicks and Barker reservoirs pool elevation during Hurricane Harvey.

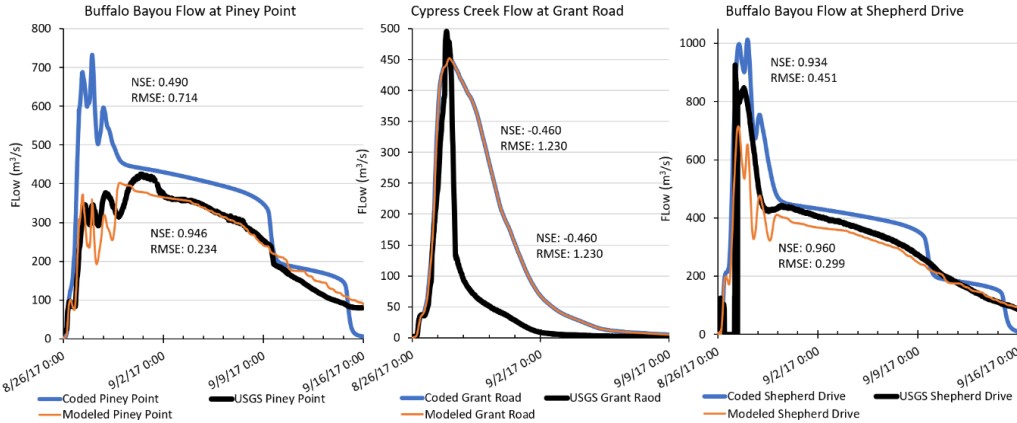

**Figure 5.** Channel flow comparisons during Hurricane Harvey.

Apart from the lack of spatially distributed rainfall, another major limitation of HEC-RAS 2D is the limited options for modeled structures. Currently, the types of hydraulic structure that could be modeled within HEC-RAS 2D are few and do not have the same computational detail of the 1D version of the model. For example, bridges could be conceptually represented in RAS 2D as a culverted weir, but the computations would lack the eddies from rounded deck piers or bank obstructions along the channel. Although inconsequential in regional watershed studies such as one presented in this paper, this limitation could pose a significant problem as the model domain decreases in size because of the compounding effects of each individual bridge or in the end results. When these issues are addressed, by either model updates, combined 1D-2D domains, or other discovered methods, this methodology will supply more accurate and detailed information for reservoir operations modeling to minimize the flooding effects in major storms. Overall, the excellent matches show that this work has laid the foundation for this modeling framework in future applications. Despite the current limitations stated, HEC-RAS 2D can accurately simulate reservoir operations in addition to the hydraulics it was designed for.

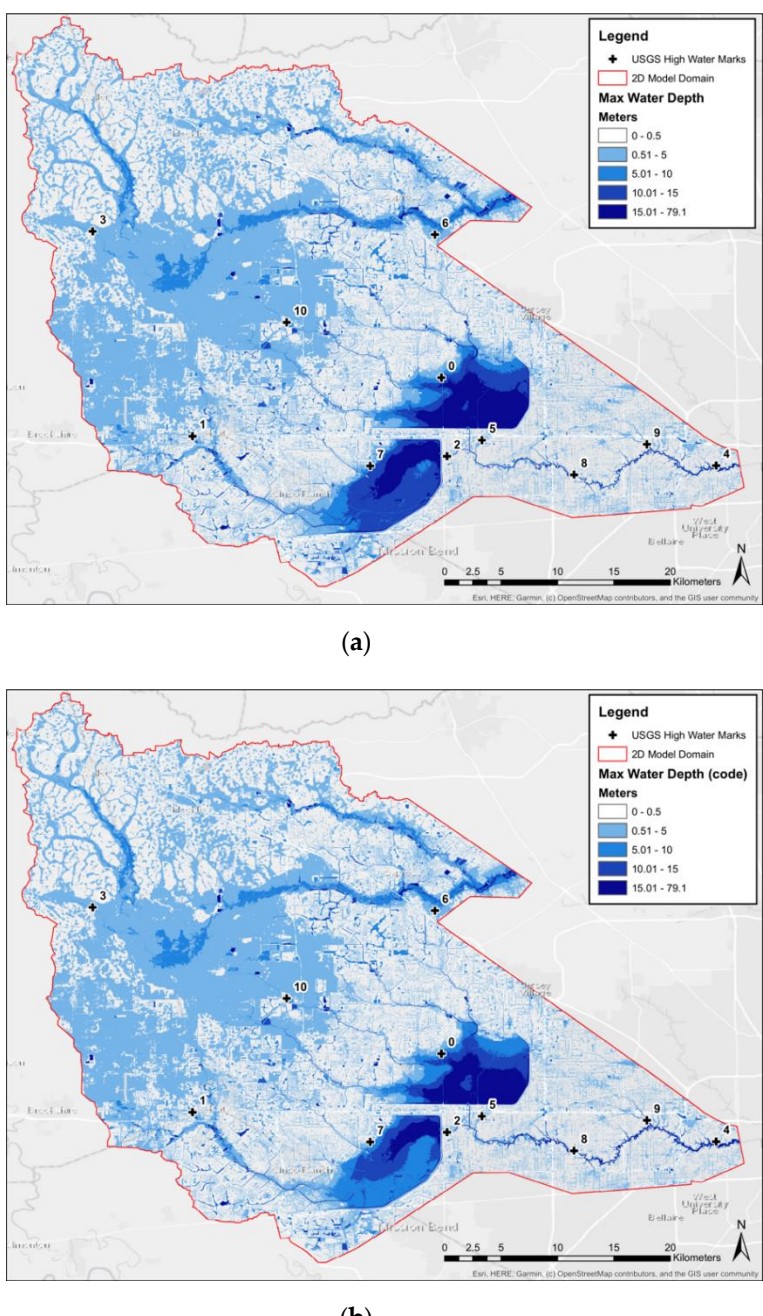

(**a**)

(**b**)

**Figure 6.** (**a**) Modeled hindcast of observed USGS operations data for Harvey floodplain extent; (**b**) Coded operations hindcast of Harvey floodplain extent.

**Table 1.** Flood depth comparison of USGS observed high water marks from Figure 6a,b.

| Point ID | USGS High Water Mark (m) | Modeled Operations Figure 6a (m) | Coded Operations Figure 6b (m) |
|---|---|---|---|
| 0 | 1.5 | 1.91 | 1.20 |
| 1 | 1.3 | 0.84 | 0.36 |
| 2 | 1.2 | 0.82 | 0.77 |
| 3 | 1.2 | 0.71 | 0.73 |
| 4 | 1.1 | 1.13 | 0.19 |
| 5 | 1.0 | 1.04 | 1.00 |
| 6 | 0.8 | 0.98 | 0.91 |
| 7 | 0.8 | 0.98 | 0.13 |
| 8 | 0.6 | 0.38 | 0.18 |
| 9 | 0.6 | 0.68 | 1.08 |
| 10 | 0.2 | 0.19 | 0.62 |

## 4. Conclusions

This study successfully demonstrates the capability of using a single model, HEC-RAS 2D, for combined hydrologic and hydraulic simulations, reservoir operations, and floodplain analysis in a hydrodynamically complex region (West Harris County, TX, USA). Using HEC-RAS 2D, the point source limitation of the conventional CWMS or WAT modeling framework is circumvented while still using free-to-use HEC software. Moreover, by using a single model to simulate all the relevant processes instead of linking separate models, the entire workflow is streamlined opening up the possibility for a reservoir WCM to account for more than just pool elevation and rate of rise in the future. Although this methodology has not yet been directly compared to CWMS or WAT, both methodologies using RAS for the hydraulics allows these conclusions to be drawn aside from computational run times.

Preemptively knowing how the emergency operations of an urban reservoir will impact flooding can mitigate the damages like those seen during Hurricane Harvey in the future. This framework can also be used for other distributed modeling functions that integrate structure operations such as dam breach modeling or regional flood protection modeling as intended with this work. This approach reduces the computational errors associated with point source connections and unaccounted gate flows seen in the traditional methodology and is more accurate in spatially distributed flow regions. These improvements over the current methods could also improve regional modeling in other related fields such as stormwater reuse, flood warning, and emergency planning. Despite the current limitations of the model (e.g., lack of infiltration loss and spatially distributed rainfall), HEC-RAS 2D is a capable and robust tool that effectively handles combined reservoir operations and flood inundation modeling.

**Supplementary Materials:** The following are available online at http://www.mdpi.com/2073-4441/12/8/2259/s1, Section S.1: *HEC-RAS 2D Sluice Gate Operation Scripts*, Figure S1: Rule operation code for Addicks reservoir sluice gate, Section S.2: *Full Coded Script of Addicks Reservoir Gate Copied from the HEC-RAS Interface*.

**Author Contributions:** M.G. student, led the modeling, hypothesis testing, and writing. A.J., research scientist, supervised the modeling and aided in writing. P.B., faculty advisor, supervised the project. All authors have read and agreed to the published version of the manuscript.

**Funding:** This research received external funding from the "Greater Houston Flood Mitigation Consortium" https://www.houstonconsortium.com/ and the "Katy Prairie Conservancy" https://www.katyprairie.org/.

**Conflicts of Interest:** The authors declare no conflict of interest.

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
