# Peer review of "Integrating Reservoir Operations and Flood Modeling with HEC-RAS 2D"

_water, doi:10.3390/w12082259_

Round 1

Reviewer 1 Report

The manuscript entitled “Integrating reservoir operations and flood modeling with HEC-RAS 2D” written by Garcia and co-authors describes a quite interesting approach to modeling reservoir control with the 2D hydrodynamic model. The control of the reservoir is done by an internal Fortran-like script written in the HEC-RAS environment. Although the approach is worth being published due to its advancement, the manuscript is written in a wrong way. A lot of elements have to be improved, e.g. purpose of the research, description of the materials, methods, discussion, and conclusions. Hence the manuscript requires intensive corrections before publication. The suggested changes are written as comments in the attached version of the manuscript.

Author Response

Attached document contains all the labeled responses. General response copied below

General Response: We really appreciate Reviewer 1’s feedback on this paper. To improve the overall study, a shift in the focus of this work was done per Reviewer 1’s comments and suggestions. First, the coded / scripted operation results were added to show and compare why the observed operations data was used for validation and calibration of the methodology. Second, clarification and additional information about the context for this work against other current models, and the need for this specific work was added in the Abstract, Introduction, Discussion, and Conclusion sections. The scripts were also added in its entirety in a format that can be easily referenced by the reader in Appendix A.2.

A second general note for a few comments made throughout the work. While we used and referenced certain data, records, and information made available by the U.S. agencies in this study,  we are unable to comment on the overall quality and accuracy of those cited data sources, but only on how they were used and performed in our particular case study. Other small comments on more detailed clarification on the data used was added throughout as asked and we agreed with the Reviewer 1 that with the new focus, this paper should be closer to a case study with what is presented than a technical note.

Reviewer 2 Report

Water MDPI

Integrating reservoir operations and flood modeling with HEC-RAS 2D

  • The authors presented a technical note on how to use USACE’s hydraulics modeling tool HEC-RAS 2D to analyze a comprehensive process that include hydrology and reservoir operations. The method described is interesting as it simplifies computational effort in estimating reservoir level and channel outlets.
  • The authors have stated the limitations associated with their current method. There is a high risk associated with flood events especially in places like Houston, TX. As a result, it will be even more prudent to include a disclaimer in the conclusion section that the results from the current method have not been compared with the conventional methods (CWMS or WAT), at least not shown here. That is, if a comparison has been made, then errors and computational costs can be quantified, and the authors can make a definitive conclusion on the performance of their method.
  • The authors should use a consistent and ordered reference format. The first reference started with [6] and a couple of lines below a direct reference (Linder 2008) is used.
  • The study period should be mentioned in the ‘Methodology’ section. Right now, one must go to result figures to see the study period. Mentioning the Hurricane Harvey event is too broad for a reader.
  • Lines 69-70: A poster presentation at a conference is not a peer-reviewed work/result that the authors can reference here. Substitute this with the appropriate reference or remove the sentence.

Author Response

Attached document contains all the labeled responses. General response copied here.

Response: A disclaimer sentence was added in the conclusion along with the addition of the coded operation runs for comparison.

Line 161: ”Although this methodology has not yet been directly compared to CWMS or WAT, the authors experience with these modeling wrappers, along with both using RAS for the hydraulics, allows for comparisons excluding the computational run times.”

Other reviewer comments suggested some major edits to multiple sections, so the scope of this paper was expanded to show an example of coded operation runs, the process of which was described in detail in the Appendix.
